# Genome-Wide Identification, Characterization, and Transcriptomic Analysis of the Cyclin Gene Family in *Brassica rapa*

**DOI:** 10.3390/ijms232214017

**Published:** 2022-11-13

**Authors:** Sumer Zulfiqar, Tiantian Zhao, Yuanming Liu, Lai Wei, Muhammad Awais Farooq, Javaria Tabusam, Jianjun Zhao, Xueping Chen, Yanhua Wang, Shuxin Xuan, Na Li, Yin Lu, Shuangxia Luo, Shuxing Shen, Aixia Gu

**Affiliations:** State Key Laboratory of North China Crop Improvement and Regulation, Key Laboratory of Vegetable Germplasm Innovation and Utilization of Hebei, Collaborative Innovation Centre of Vegetable Industry in Hebei, College of Horticulture, Hebei Agricultural University, Baoding 071000, China

**Keywords:** cyclin gene family, *Brassica rapa*, leaf size, SNP, transcriptomic analysis, genome-wide analysis

## Abstract

Cyclins are involved in cell division and proliferation by activating enzymes required for the cell cycle progression. Our genome-wide analysis identified 76 cyclin genes in *Brassica rapa*, which were divided into nine different types (A-, B-, C-, D-, H-, L-, P-, T-, and SDS-type). Cyclin genes were unevenly scattered on all chromosomes, with a maximum of 10 on A08 and a minimum of 2 on A04. The gene structure and conserved motif analysis showed that the cyclins which belonged to the same type or subgroup have a comparable intron/exon pattern or motif. A total of 14 collinear gene pairs suggested that the *B. rapa* cyclin genes experienced a mass of segmental duplication. The Ka/Ks analysis revealed that the *Brcyclin* gene family has undergone an extensive purifying pressure. By analyzing the cis-elements in the promoters, we identified 11 cis-elements and five of them are related to the hormone response. We observed 48 potential miRNAs targeting 44 *Brcyclin* genes, which highlighted the involvement of miRNAs in the regulation of cyclin genes. An association analysis between the leaf size and SNPs in mutants and a transcriptome analysis of two Chinese cabbage-cabbage translocation lines also showed that the *Brcyclin* gene family was involved in the development of the leaves. The functional characterization of the *B. rapa* cyclin gene family will provide the foundation for future physiological and genetic studies in the regulation of leaf growth.

## 1. Introduction

The cyclin protein family controls cell progression (division and expansion) by interacting with cyclin-dependent kinases (CDKs), which regulate the cell cycle in plants [1,2]. Cyclin, first identified in the soybean [3], plays an important role in the cell cycle. Cyclins are classified on the basis of their amino acid sequence and the point or phase in the cell cycle during which they activate their CDK partners. The phase has been divided into two categories, M- and G1-phases [4]. The former includes A- and B-type cyclins, which aid in the entry of cells into the M-phase through the nuclear division (mitosis), followed by the cytoplasmic division (cytokinesis), whereas the latter includes cyclins C-, D-, and E-type, that become active towards the end of the G1-phase and are responsible for ushering the cell into the S-phase. The G1 phase also serves as a critical checkpoint for cells, determining whether the cell will divide or remain stalled in the G1 phase of the cell cycle. Typically, cyclin contains a conserved region known as the cyclin core, which consists of two domains: the first one is cyclin N, known as the cyclin box, and is more conserved than the second domain, which is cyclin C [5]. Some cyclins only contain a cyclin-N domain but have no cyclin-C domain [6], which indicates the cyclin-C domain isn’t crucial for its function. In *Arabidopsis*, 50 cyclins have been classified into ten types, based on function and sequence analysis: A- to D-types, H-, L-, T-, U-, SDS-, and J18-types [7]. Most cyclins are expressed and function during different phases of the cell cycle. Plants accumulate different A-type cyclins from the early G1 to S phases and until the middle M phase of the cell cycle [8]. Overexpression of the tobacco *CYCA3;2* cyclin caused ectopic cell division and delayed differentiation, which was linked to an increase in S phase-specific gene expression and the *CYCA3;2*-associated CDK activity [9]. These findings imply that A-type cyclins are expressed throughout the cell cycle and may have a wide range of functions in plants. The interphase in mitosis is the longest period of the cell cycle, during which the cell is not dividing but preparing itself for division. In the intra-G2-phase and the intra-M-phase control, B-type cyclins are expressed for a short time [10,11]. The overexpression of the dominant-negative *CDKB1;1* results in an early exit from the M phase, which increases the ploidy level in the leaf [12]. CDKBs play crucial roles in the mitotic-to-endocycle transition and also appear to be activated during the DNA damage response in plants [13].

D-type cyclins have a high expression and are one of the largest plant-specific groups. They are divided into seven sub-groups (CYCD1 to CYCD7) [7] and interact with CDKs to participate in the G1-to-S phase with A-type cyclins. The D-type cyclin *CYCD3;1* is, for example, a limiting factor in the G1-to-S phase transition [14].

D-type cyclins, *CYCD3;1* and *CYCD2;1*, are hypothesized to function as important switches in the initiation of hormonal effects. For example, brassinosteroids (BRs) accentuate cell division by upregulating the expression of CYCD3 through a process that necessitates de novo protein synthesis [15]. In the *Arabidopsis* leaf, a *triple cycd3;1–3* loss of function results in a lower cytokinin (CK) response, along with fewer cells [4,16,17,18]. The ABP1 (AUXIN BINDING PROTEIN1) suppression negatively affects the CYCD transcript levels, resulting in a decreased cell division in the leaf. Similarly, in tobacco plants, the overexpression of *A. thaliana*’s dominant-negative *CDKA;1*, slowed cell division, resulting in fewer but larger cells with a decreased leaf area [19]. Additionally, apart from CYCA, CYCB, and CYCD-type cyclins, other cyclins have not been thoroughly explored in plants, in terms of their function.

Till now, the cyclin gene family has been explored in a variety of plant species, except for *Brassica rapa* [20,21]. *Brassica rapa* is a valuable genetic resource for crop breeding and an ideal material for plant development research, due to its high genetic diversity and phenotypic variation. Its leafy structure has given it tremendous economic and nutritional value, and the cyclin gene family is crucial for the growth and development of leaves. The suppression of CYCA, CYCB, and CYCD transcription levels may cause a cell cycle arrest and a reduction in the CDK activity [22], which alters leaf development [23]. Therefore, a genome-wide analysis was conducted to find the cyclin gene family in *Brassica rapa*. The identified 76 cyclin genes were further analyzed for protein sequences, gene characterization, chromosomal localization, phylogeny, collinearity, synteny, 3D protein structure, micro-RNA targeting gene analysis, promoter cis-elements, and the expression analysis through the transcriptomic data. These bioinformatic analyses will provide novel insights into the evolutionary history and functional divergence of the cyclin gene family in *Brassica rapa*.

## 2. Results

### 2.1. Identification of the Cyclin Gene Family in Brassica rapa

To identify the cyclin genes in the *Brassica rapa* genome, BLASTP was used on the Brassica database (http://brassicadb.cn/#/, accessed on 1 September 2022) by using the 10 *Arabidopsis* genes as a query sequence. As a result, we discovered 181, 196, 16, 222, 7, 14, 17, 37, and 49 members in the A-, B-, C-, D-, H-, L-, T-, SDS-, and P-types, respectively. We compared all of the identified genes and found that the A-, B-, D-, and SDS-types shared the same cyclin genes (repetition of the same transcripts), but some of them were different, whereas the same scenario was found with the C-, H, L-, and T-types. However, the P-type was found to contain different genes (no repetition of the transcript IDs was found). Repeated transcripts were excluded, and the actual remaining 76 distinct cyclin genes in the entire genome were kept for further analysis. We decided that the candidates containing at least one cyclin-N domain should be considered as “true” cyclins [7]. Of the 76 unique members, 59 genes contain both N- and C- domains, and only 17 genes contain only the N-domain (Appendix A). The length of the *B. rapa* cyclin proteins ranges from 113 to 1229 amino acids (aa), with an average of 400 aa. The smallest *B. rapa* cyclin protein, *B. rapa* CYC-P-76 (113 aa), and the biggest *B. rapa* cyclin protein, B. rapaCYC-C-51 (1229 aa), contain both N- and C- domains. All predicted genes and sequencing IDs, gene descriptions, chromosome locations, instability index, protein length, molecular weight, PI, and GRAVY have been presented (Table 1).

The putative cyclin genes were localized on all chromosomes, but the general distribution was mostly unequal, with a maximum of 10 on A08 and a minimum of 2 on A04 (Figure 1).

### 2.2. Cyclin Gene Structure and the Conserved Domain and Motif Analysis

A gene’s structure (exon/intron) distribution is strongly linked to its function. The *B. rapa* evolutionary tree showed that the maximum cyclin members within a group contained the same gene structure (Figure 2a,b). Both cyclin domains N- and C- were present in all members of the A-, B-, and D- types (plant-specific) of the *B. rapa* cyclin genes (Appendix A). Using the MEME algorithm, 10 types of motifs and their distribution in the *B. rapa* cyclins were predicted to understand the distribution of the motifs in cyclin proteins and their function. The study revealed that cyclins belonging to the same group have comparable motifs. Furthermore, all probable *B. rapa* cyclins were grouped into five major groups (Groups (I) red, (II) purple, (III) blue, (IV) yellow, and (V) green). Group (I) contained 16 *Brcyclin* members; out of them, four members contained similar conserved motifs, 11 members contained two similar conserved motifs, and only one member contained three similar motifs. Group (II) contained varying numbers of motifs; with 15 *Brcyclin* members, six have three motifs, one has two, and eight have four conserved motifs and one variable motif. Group (III) consisted of 12 *Brcyclin* members; 11 have four conserved motifs, and one has three conserved motifs. Group (IV) consisted of 14 *Brcyclin* members; six contained eight conserved motifs, seven members contained nine conserved motifs, and one member has three conserved motifs. Group (V) contained 19 members; all have nine conserved motifs (Figure 2b).

### 2.3. Comparative Phylogenetic Analysis and Classification of the B. rapa Cyclin Gene

A phylogenetic study was performed using the maximum likelihood tree approach to understand the evolutionary relationship of the *B. rapa* cyclins with the Brassicaceae species: *B. napus* (Bna), *B. oleracea* (Bol), *B. juncea* (Bju), *B. nigra* (Bni), and *A. thaliana* (At). The tree was built using 76, 27, 25, 34, 38, and 49 cyclin genes from *Brassica rapa, Brassica napus, Brassica oleracea, Brassica juncea, Brassica nigra*, and *Arabidopsis*, respectively. The tree was divided into six clusters (I-VI) (Figure 2). According to our results, Group I comprised 58 cyclin members, Group II comprised 37, Group III comprised 38, Group IV comprised 47, Group V comprised 30, and Group VI comprised 50 cyclin members (Figure 3). In addition, it is established that the *B. rapa* cyclins evolved closely with other Brassicaceae species. Furthermore, CYC-A was one of the major types in *Brassica rapa*, with 15 described genes, followed by CYC-D with seven defined and five uncharacterized genes, and the CYC-B type cyclin with nine genes.

### 2.4. Duplication Event in the Brassica rapa Cyclins and Synteny Analysis

The tandem and segmental duplication advance the development and progression of plant genomes by promoting new gene family members. Through polyploidy, segmental duplications result in gene duplication, whereas tandem duplications result from the crossing over of shorter fragments [23,24,25]. We used BlastP searches to comprehend the tandem and segmental gene duplication in cyclins. Out of 76 *Brcyclin* genes, we found 14 collinear gene pairs, indicating that all pairs were segmentally duplicated, and no tandem duplicated pairs were found (Figure 4a). These findings show that segmental duplication played an important role in the expansion of the *Brcyclin* family genes.

A synteny analysis showed that the *B. rapa* cyclins displayed the most collinearity with *B. napus*, *B. juncea*, and *B. carinata*, followed by *B. oleracea*, *B. nigra*, and *A. thaliana* (Figure 4b). A total of 75 *B. rapa* cyclin genes were found to syntenic with *B. napus*, *B. juncea*, and 69 with *B. carinata*, 67 with *B. oleracea*, and *B. nigra*, while 62 with *A. thaliana*. The study revealed that *B. rapa* and the other six Brassicaceae species had strong orthologs of the cyclin genes. Several homologs of other Brassicaceae crops maintained a syntenic relationship with the *B. rapa* cyclin genes, implying that whole genome duplication, along with segmental repetition, played a crucial role in the *Brcyclin* gene family evolution. Furthermore, we determined the nonsynonymous (Ka) and synonymous (Ks) ratios for each pair of duplicated *Brcyclins* under a selective sweep (Table 2). All *Brcyclin* pairs had a Ka/Ks ratio of less than 1, indicating that the *Brcyclin* gene family has undergone extensive purifying selection pressure.

### 2.5. Cis-Elements in the Promoters of the Brassica rapa Cyclin Genes

The 2.0 kb upstream sequences from the translation start sites of the cyclin genes were submitted to the PlantCARE database to detect the cis-elements in the promoters of the *Brcyclin* genes, to study the potential regulatory mechanisms in the biological processes, particularly in plant hormones, meristem development, cell cycle regulation, and pathogen infection. Results showed 11 cis-elements (Appendix A), among them five were hormone-related response elements, i.e., abscisic acid (ABA), auxin, methyl jasmonate (MeJA), gibberellin (GA), and salicylic acid (SA), implying that these genes may respond to phytohormones (Figure 5a and Appendix A). *Brcyclin* genes *CYCA2;3*, *CYCA2;4*, *CYCD3*, *CYCD3;1*, and some uncharacterized cyclin genes, are hormone-correlated responsive elements that play a significant role in plant growth and development (Table 1). Furthermore, the stress responsive cis-elements were also found, which include drought, low temperature-responsive elements, defense, stress, light, and circadian (Figure 5a,b). The maximum elements were found for the light response and the auxin hormone response, and the minimum elements were found for the circadian responsive elements in all of the promoters of *Brcyclin* genes (Figure 5b), implying that *Brcyclin* genes may respond to stress stimuli as well and have a wide range of functions during plant development.

### 2.6. 3D Protein Structure of the Cyclin Genes in Brassica rapa

Proteins are important molecules that play a significant role in different biological processes. The function of a protein is largely dependent on its 3D structure, so protein structure prediction or modeling is very significant. The 3D structure of a protein offers insight into how it functions and how it might be utilized, to control or modify it. It also enables the understanding of several biological interactions, and the prediction of which molecules will bind to that protein. In this study, 3D protein structures were predicted to understand the attributes of proteins and facilitate comparative studies of them. We inferred that all 76 *Brcyclin* genes have a conserved 3-dimensional structure with alpha (α)_ helix and beta (β) sheets after constructing the 3D protein structures (Figure 6), because proteins with similar structures often have similar functions. The protein structures were modelled via the c2khoA template at a confidence level of 100%. On the basis of the protein structure guide design, binding ligands can be determined, that are essential to bind with the external or internal residues. In addition, the cyclin protein family’s tertiary structure predicts the physical connections, different functional roles, independent divergence sequences, and conserved biological features.

### 2.7. Genome-Wide Analysis of the miRNA Targeting Cyclin Genes in Brassica rapa

miRNAs play a critical role in plant growth and development, including the signaling pathways [26] and organ morphology [27], by silencing and upregulating their target genes. The variability of the DNA sequences at the miRNA target sites alters the complementation between miRNAs and their target genes, which influences the target gene expression levels and the traits relevant to these genes (Appendix A) [28]. “MicroRNA-targeting a specific gene family analysis” conveys the positional information needed to generate the developmental patterns of a specific genetic regulator [29]. We observed 48 potential miRNAs targeting 44 *Brcyclin* genes, which highlighted the involvement of miRNAs in the regulation of the cyclin genes (Figure 7). *B. rapaCYC-17* was targeted by a maximum of 14 miRNAs, followed by *B. rapaCYC-10*, which was targeted by 12 miRNAs, *B. rapaCYC-11*, and *B. rapaCYC-18*, were targeted by 10 miRNAs (Appendix A and Figure 7).

### 2.8. Functional Annotation Analysis of the Cyclin Genes in Brassica rapa

The GO annotation and enrichment analysis were carried out to understand the function of the *Brcyclin* genes in biological process (BP), molecular function (MF), and cellular component (CC). The GO annotation was only detected in the P-type cyclin, and the annotations were only found in BP and MF (Appendix A). In the GO-MF, only one highly enriched term **(GO:0000079)** was found, which demonstrated its involvement in protein kinase binding activity, cell growth and proliferation, as well as the initiation and modulation of the immunological responses. Similarly, in the GO-BP, only one highly enriched term was anticipated **(GO:0019901)** which was found to be involved in the control of the cyclin-dependent protein serine/threonine kinase activity. These GO enrichment results suggested that the *Brcyclin* genes play a pivotal role in growth and development.

### 2.9. The Functional Characterization of the Brassica rapa Cyclin Genes by the SNP Analysis

Out of the 76 cyclin genes, 74 genes showed 523 SNPs in the CCEMD, which was created by our research team (Appendix A), and 61 cyclin genes had 138 non-synonymous SNPs in the body region of the gene (Appendix A) [30]. These 138 non-synonymous mutations were attained from 106 mutant plants, and among them, 21 mutant plants showed altered leaf size, as compared to the wild-type plants (Appendix A).

### 2.10. Expression Profiles of the Brassica rapa Cyclin Genes by a Transcriptome Analysis

According to our transcriptome results between the normal and small-leaf plants, out of 76 *Brcyclin* genes, a total of 35 expressed genes were identified in the Chinese cabbage-cabbage 16-1 translocation line. There were 13 genes with log2FC| ≥ 1, FDR ≤ 0.01 and all were downregulated in 16-1S plants, compared with 16-1N plants (Figure 8 and Appendix A), including six CYCA genes, two CYCB genes, two CYCD genes, and three uncharacterized genes. (Figure 8, Appendix A). It suggests that the CYCA genes could play a significant role in regulating the size of 16-1 plant leaves. Similarly, in the qdh-263 Chinese cabbage-cabbage translocation line, 31 expressed genes were identified out of 76 *Brcyclin* genes, with an FPKM value ≥ 3.17, of them 17 had |log2FC| ≥ 1 and FDR ≤ 0.01 (Figure 9 and Appendix A). In our transcriptomic results, *CYCA2;3* (*BraA09g058650.3C*) and *CYCA3;1* were downregulated in the 16-1S plants. The downregulation of these genes may be the reason for the small leaf size of the 16-1S plants because *CYCA2;3* acts as a major regulator of the ploidy levels in *Arabidopsis* endoreduplication by negatively regulating the endocycles [11]. *CYCA2;3* alters the ploidy levels and endocycles, which affect leaf development. Corneillie, De Storme et al. 2019 have reported the morphometric analysis of leaves in which the polyploidy influences the epidermal pavement cells, resulting in larger cell sizes and fewer cells per leaf blade, as the ploidy rises [31]. Further studies with reference to the ploidy level and its effects on leaf size can bring new perspectives to leaf development. The genes *CYCB2;3* (*BraA08g027760.3C*) and *CYCD3;1* (*BraA08g16120.3C*, *BraA01g004130.3C*) play an important role in the cell cycle and cell division, respectively [32]. The downregulation of these genes affects the growth and development of plant organs, which is consistent with our transcriptomic results.

Furthermore, there are a few uncharacterized genes in the cyclin gene family, i.e., *BraA07g016670.3C, BraA07g027470.3C*, and *BraA07g039870.3C* (core cell cycle genes), according to the Brassica database http://brassicadb.cn/#/Annotations/, accessed on 1 September 2022 (Table 3 and Appendix A), which need to be studied and are probably involved in the regulation of leaf size. A similar cyclin gene family behavior was observed in the qdh-263 translocation line. The expression of all plant-specific cyclin types A-, B-, and D- was downregulated (Figure 9). These results imply that in our accessions, the small leaf size might be due to the poor functioning of the cell cycle genes. It also suggests that in both translocation lines, the pathway of the *Brcyclin* genes in the regulation of the leaf size is same.

## 3. Discussion

In plants, the cyclin-dependent kinases (CDKs) regulate cell division, plant growth, and development, including leaf development and branch formation. The CDK activity is regulated by the kinase’s interaction with a regulatory cyclin component, which regulates the timing of CDK activation, the complex’s subcellular location, and its substrate specificity. Cyclins are involved in virtually every step of the mitotic process by interacting with CDKs and other proteins [33].

### 3.1. Identification of the Cyclin Gene Family and Its Structure in Brassica rapa

In plant genomes, the number of cyclin genes varies; we found 76 cyclin genes in the *Brassica rapa* genome of nine different types (A-, B-, C-, D-, H-, L-, P-, T-, and SDS-types), and the A-type was the largest one. The rice genome has been found to include 49 cyclins, which are divided into nine types: A-, B-, D-, F-, H-, L-, P-, T-, and SDS-types [34]. The maize genome contains 59 cyclins that can only be divided into six types: A-, B-, D-, F-, T-, and SDS-type cyclins [15]. The poplar genome contains 45 cyclins that can be categorized into seven types: A-, B-, C-, D-, Q-, T-, and Z-types [35]. There are 50 cyclins in the *Arabidopsis* genome, which are divided into ten types: A-, B-, C-, D-, H-, L-, U-, T-, J18-, and SDS-types [7]. 52 cyclin genes in the tomato genome have been identified, which are also classified into 10 types: A-, B-, C-, D-, H-, L-, P-, T-, J18-, and SDS-types, and legumes share the same categories as *Arabidopsis*. Previously, it was discovered that cyclin types A-, B-, D-, and U- were further divided into 3, 3, 7, and 4 subgroups, respectively, while other types C-, H-, L-, SDS-, and T- were not divided into any other subgroups [20]. All plant genomes have A-, D-, and T-type cyclin genes, so these three types are more conserved. The structural investigation of the *B. rapa* cyclins demonstrates their remarkable conservation and distinguishing characteristics (Figure 2a).

### 3.2. Segmental Duplication Event of the Brassica rapa Cyclins

Gene duplication is a major driving force in the evolution of genomes and genetic systems [36]. The tandem and segmental duplications cause the substantial variations in the family size, as observed in most gene families [25]. Studies on the *Arabidopsis* genome have featured a high number of large segmental duplications that came from ongoing polyploidy events and have been scrambled by chromosomal rearrangements [34]. The co-linearity of the cyclin genes among the A, B, and C genomes indicates that these genomes are the progenitors of a common ancestor and that, over the course of evolution, these genes have remained conserved, due to their utility in the cell cycle progression. Moreover, the collinearity events of the *B. rapa* cyclin genes with all other Brassica crops, showed their similar evolutionary progenitor. Throughout evolution, it seems that cyclin gene duplication and subsequent expansion have occurred frequently. These discoveries shed light on *Brassica rapa*’s genome evolution, which suggests that all segmental duplicated cyclin genes have comparable or overlapping activities, despite their striking variances in amino acid sequences.

### 3.3. MicroRNAs Targeting the Cyclin Genes in Brassica rapa

MicroRNAs (miRNAs), a family of single-stranded, non-coding RNAs, have been discovered to have a role in post-transcriptional gene regulation [37,38]. Numerous miRNAs have been identified in Chinese cabbage through genome-wide studies in recent years [39]. In our study, 48 putative miRNAs targeting 44 *Brcyclin* genes have been identified (Figure 7). These findings suggest that Br-miRNAs may play a critical role in protecting the plant from a variety of stresses, by changing the transcript level of the *cyclin* genes in *Brassica rapa*.

### 3.4. Expression Profiles of the Brassica rapa Cyclin Genes by Analyzing the Transcriptomic Data

Out of 76 *Brcyclin* genes, a total of 13 differentially expressed genes were identified in 16-1S and 16-1N, and 17 differentially expressed genes were identified in qdh-263S and qdh-263N. All of them were downregulated in 16-1S and qdh-263S, compared with 16-1N and qdh-263N.

In plants, A-type cyclins are classified into three subtypes, namely, CYCA1, CYCA2, and CYCA3. These genes showed different expression levels in 16-1N and 16-1S accessions (Figure 8). The protein CYCA1;1 is a regulatory subunit and CDC2OS-3 is a catalytic subunit of the cyclin-dependent protein kinases (CDKs), which are central to the cell cycle regulation. Both genes are regulated by gibberellic acid in a coordinated manner, with transcripts accumulating in the G2 phase, prior to the B2-type mitotic cyclins [40], suggesting a distinct role in regulating the G2/M phase progression. *CYCA3;2* is a distinct member of the G1 cyclin family that plays an important role in the meristematic tissues.

The gene *CYCB2;3* (*BraA08g027760.3C*) is the cyclin-dependent protein kinase regulator activity, which is involved in the regulation of the cell cycle. The downregulation of this gene resulted in decreased leaf size [32], which is consistent with our transcriptomic results. Therefore, we speculated that *CYCB2;3* might be involved in the regulation of the leaf size in our accessions.

*CYCD3;1* (*BraA08g16120.3C*, *BraA01g004130.3C*) plays a key role in integrating cell division in the leaf and lateral organ development [41]. The *CYCD3;1* expression is increased by the cytokinin, which regulates a variety of developmental programs, including the shoot regeneration, leaf development, greening, and cell cycle stimulation. A mutation in *CYCD3* decreases the shoot meristem activity and leads to a reduced cytokinin response [18]. It showed that the *CYCD3* activity is critical for the cell quantity in growing lateral organs and mediates the cytokinin effects in the apical growth and development. These findings suggest that the small leaf size in our accessions may be due to a cell cycle malfunction; the dysfunction of the cell cycle may be caused by the cyclin genes’ aberrant behavior. In the future, more research will be required to map these regulators in the bigger network of leaf growth regulation.

## 4. Material and Method

### 4.1. Identification and Characterization of the Cyclin Gene Family

A total of 76 cyclin genes (http://brassicadb.cn/#/BLASTP/, accessed on 1 September 2022) were retrieved from the genome assembly v3.0, on the basis of the e-value < 0.01, by using the 10 *Arabidopsis* cyclin genes (At1g44110, At4g37490, At5g48640, At1g70210, At5g27620, At2g26430, At1g35440, At1g14750, At3g21870, and At2g01820) as a query sequence, which were taken from the *A. thaliana* genome (https://www.arabidopsis.org/about/citingtair.jsp, accessed on 1 September 2022) database. Further screening was done on the basis of the PF value (**PF00134**, cyclin, N-terminal, **PF02984**, cyclin, C-terminal) (http://pfam.xfam.org/, accessed on 1 September 2022) *B. rapa cyclin* proteins were further characterized by determining the molecular weight, the number of amino acids, the isoelectric point, and GRAVY through the ProtParm tool (ExPASy-ProtParam tool). The protein sequences were uploaded to the online database of Wolf PSORT for the prediction of the subcellular localization (https://wolfpsort.hgc.jp/, accessed on 1 September 2022) [42]. To visualize the chromosomal location of all obtained cyclin genes, the “Show Genes on Chromosome” feature of the TB-tools software was used.

### 4.2. Comparative Phylogenetic Analysis and Classification of the Brassica rapa Cyclin Gene

The protein sequences of the cyclin genes in *Arabidopsis, B. nigra, B. junacea, B. olearcea*, *B. carinata*, and *B. napus* were used to generate the phylogenetic tree, using MEGA (V.11 https://www.megasoftware.net/, accessed on 1 September 2022) software. The sequences were multiple aligned and employed the neighbor-joining tree method with 1000 bootstrap replicates [43].

### 4.3. Tandem Duplication Events of the Brassica rapa Cyclins and the Synteny Analysis

The duplicated genes were retrieved, using the Blast, MCScanX, and Advance Circos features of TB-tools (https://github.com/CJ-Chen/TBtools, accessed on 1 September 2022). For the purity selection pressure of the duplicated genes, the Ka/Ks values were calculated, using the Ka/Ks Calculator in TB-tools, and the time million years ago (TMYA) was calculated with the help of previous studies in *Brassica rapa* [44,45]. The synteny relationships of *B. rapa* with *B. oleracea*, *B. napus*, *B. junacaea*, *B. carinata*, *B. nigra*, and *A. thaliana* were developed by using MCScanX to get the collinearity files, which were then used to create the dual synteny graphics through TB-tools (https://github.com/CJ-Chen/TBtools, accessed on 1 September 2022) [46].

### 4.4. Cis-Elements in the Promoters of the Brassica rapa Cyclin Genes

The putative cis-elements of all 76 *Brcyclin* were classified by downloading the 2000 bp upstream of the start codon from the BRAD database. For further classification, the PlantCARE web-based tool (http://bioinformatics.psb.ugent.be/webtools/plantcare/html/, accessed on 1 September 2022) was employed, and the findings were presented using TB-tools.

### 4.5. 3D Protein Structure of the Cyclin Gene in Brassica rapa

The 3-dimensional structure of the *Brcyclin* proteins was digitally constructed, using the available web-based software PHYRE2 (PHYRE2 Protein Fold Recognition Server) (ic.ac.uk, accessed on 1 September 2022).

### 4.6. Cyclin Gene Structure, Conserved Domain, and the Motif Analysis

The protein conserved domain was identified using the NCBI conserved domain online server (www.ncbi.nlm.nih.gov, accessed on 1 September 2022), and the motif analysis was performed, using the MEME Suite (meme-suite.org, accessed on 1 September 2022). The gene structure (exon and intron) was predicted by the Gene Structure Display Server (GSDS) (http://gsds.cbi.pku.edu.cn/, accessed on 1 September 2022), using the CDS and genomic sequences of 76 *Brcyclin* genes.

### 4.7. Genome-Wide Analysis of the miRNA Targeting the Cyclin Genes in Brassica rapa

The CDS sequences of *Brcyclin* were utilized to find the microRNAs, targeting the *Brcyclin* genes through the psRNATarget database (http://plantgrn.noble.org/psRNATarget, accessed on 1 September 2022), and the interaction was further predicted by a graphical illustration.

### 4.8. Functional Annotation Analysis of the Brcyclin genes

To investigate the functional properties, Gene Ontology (GO) annotation was used by accessing the Brassica database (http://brassicadb.cn, accessed on 1 September 2022). The transcript IDs of the *Brcyclin* genes were used to find the annotation of every gene.

### 4.9. Analyzing the Transcriptomic Data and the Single Nucleotide Polymorphisms to Determine the Functional Characterization of the Brassica rapa Cyclin Genes

#### 4.9.1. Plant Growth and RNA Sample Collection

The Chinese cabbage-cabbage translocation lines 16-1 and qdh-263 were used (natural microspore-derived doubled haploid plants), which were developed by the Key Laboratory of Vegetable Germplasm Innovation and Utilization of Hebei from an introgression line, containing the chromosome segments of cabbage (*B. oleracea* var. *capitata*) in a Chinese cabbage (*B. rapa* ssp. *pekinensis*) background. The plant materials include the autotetraploid Chinese cabbage 4XL (AAAA, 2n = 40) and the inbred line cabbage (11-1; CC, 2n = 18) [47]. Heterotriploids (genome AAC, 2n = 29) were produced by crossing 4XL and 11-1, then the monosomic alien addition lines of the Chinese cabbage-cabbage (AA + 1C, 2n = 20 + 1 = 21) were obtained through selfing after backcrossing AAC and a Chinese cabbage (85-1, AA, 2n = 20), the translocation lines were developed after the isolated microspore culture (Figure 10) [48]. In our study, two phenotypes, including normal (16-1N, qdh263N) and small (16-1S, qdh263S) leaves, were present in the self-breeding offspring of 16-1 and qdh-263 (Figure 11). For the transcriptomic analysis, the seeds were grown in a greenhouse at Hebei Agricultural University for six weeks. The fifth leaves were sampled from at least three plants with similar characteristics and phenotypes. Following the sampling, the tissues were quickly frozen in liquid nitrogen and stored at −70 °C until the RNA isolation. Three technical replicates were taken from each morphotype.

#### 4.9.2. RNA-Seq Library Construction and Sequencing

The RNA sample preparations employed a total of 3 g RNA per sample, as the input material. A NEBNext^®^ Ultra™ RNA Library Prep Kit for Illumina^®^ was used to create our RNA-Seq libraries (NEB, Ipswich, MA, USA). Using poly-T oligo-attached magnetic beads, the mRNA was extracted from the total RNA. The NEB proprietary fragmentation buffer was used to fragment the divalent cations at a high temperature (5X). On an Illumina HiSeq 4000 platform, the RNA-Seq libraries were sequenced to obtain the 150 bp paired-end reads.

#### 4.9.3. Differential Expression and Cluster Analysis

The DESeq (2012) R package was used to perform the differential expression analysis. Differentially expressed genes (DEGs) were discovered using the hierarchical cluster analysis. In each pairwise comparison, the DEGs with a false discovery rate (FDR) ≤ 0.01, |log2 fold change| ≥ 1 were filtered.

#### 4.9.4. Single Nucleotide Polymorphisms (SNP) Analysis

To determine the function of the cyclin genes, we collected SNPs of the *Brcyclin* genes in lines from the Chinese cabbage EMS-induced Mutant Database (CCEMD) (http://www.bioinformaticslab.cn/EMSmutation/home/, accessed on 1 September 2022) [30] and analyzed the relationship between the phenotypic changes and the SNPs in these genes.

## 5. Conclusions

In conclusion, a total of 76 cyclin genes were identified in the *Brassica rapa* genome.

All *Brcyclin* genes were found to possess similar gene structures and conserved motifs, which showed a close evolutionary and syntenic relationship with *Arabidopsis* and all other Brassica crops. All *Brcyclin* genes were highly responsive to hormones and abiotic stress. The segmental duplication occurred frequently through the evolution of the Brassica genome. The functional analysis discovered that the *Brcyclin* gene family plays an important role in plant growth and leaf development by regulating the cell cycle. These discoveries will lay the basis for studying the roles of cyclin genes in the *Brassica rapa* developmental processes by using different functional confirmation methods, such as overexpression, RNAi, and genome editing.

## Figures and Tables

**Figure 1 ijms-23-14017-f001:**
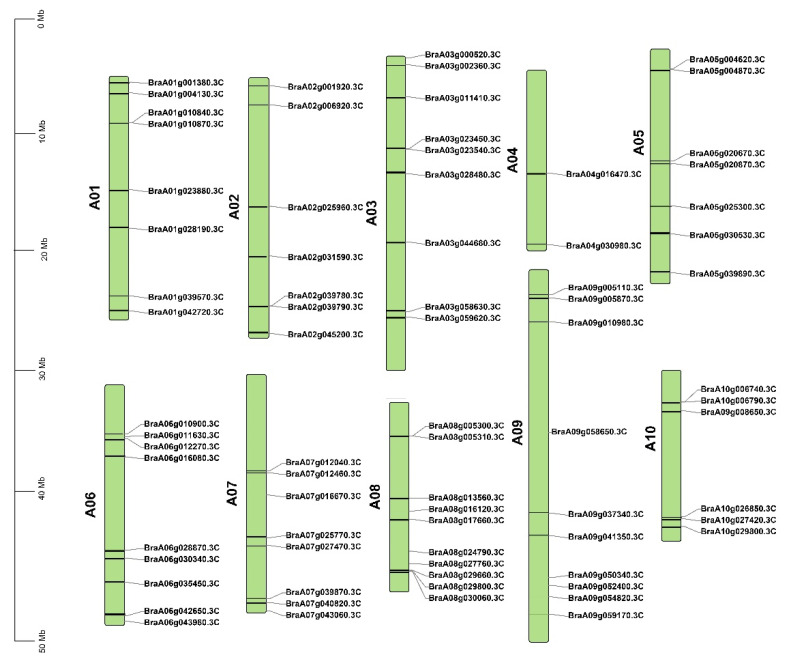
Chromosomal map of *Brassica rapa* with the distribution of the *Brcyclin* gene family member on the cabbage chromosome; the left scale indicates the size of each chromosome.

**Figure 2 ijms-23-14017-f002:**
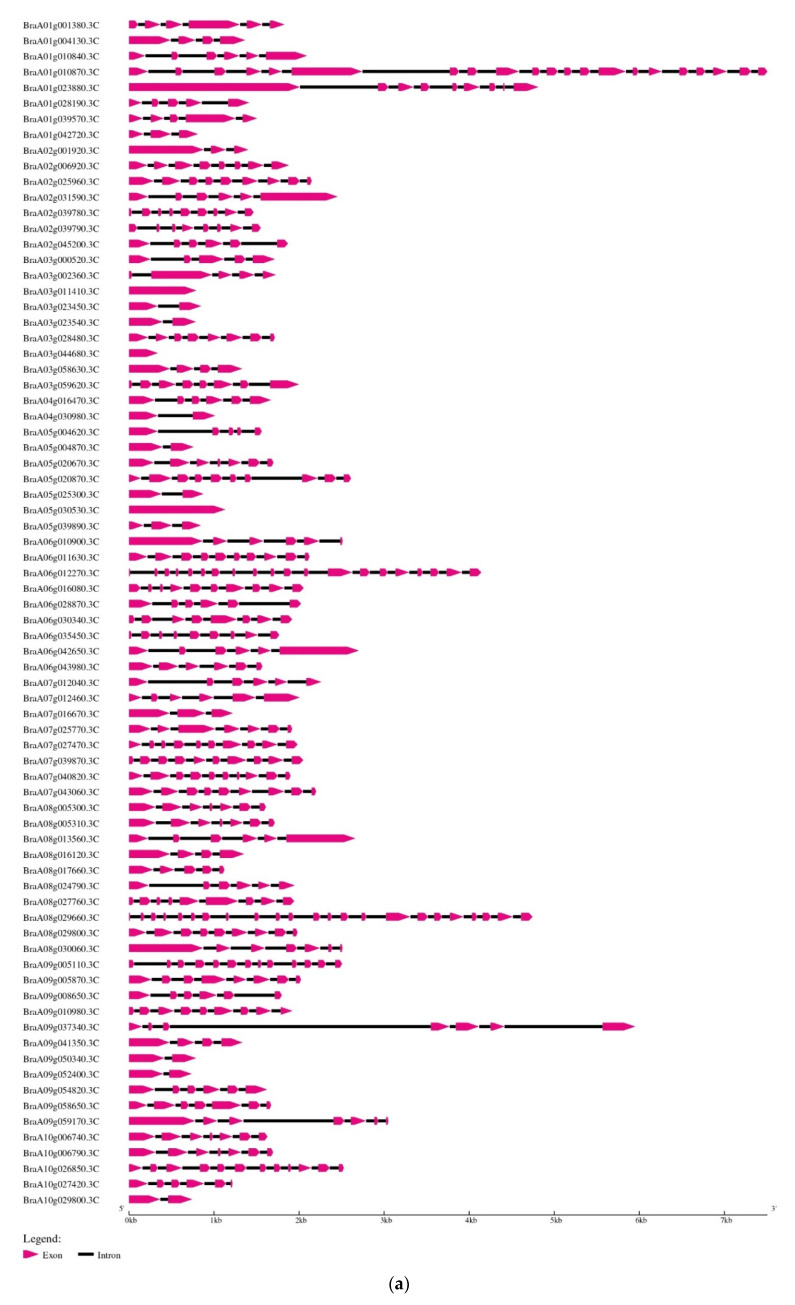
Graphical representation of the *Brcyclin* gene (**a**) Structure and (**b**) Conserved domain structures identified in the *Brcyclins*.

**Figure 3 ijms-23-14017-f003:**
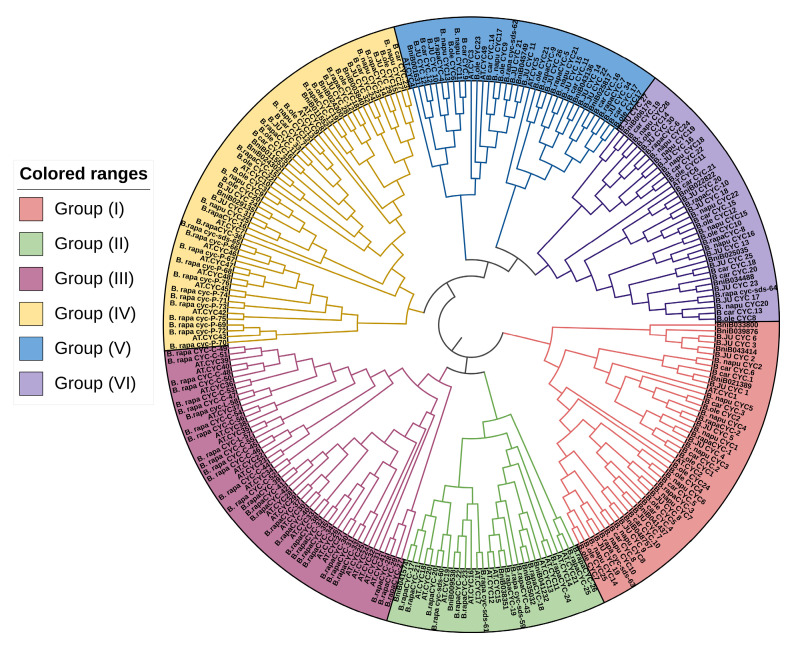
Phylogenetic tree analysis of the cyclins retrieved in *Brassica rapa*, *Arabidopsis thaliana*, *Brassica napus*, *Brassica oleracea*, *Brassica juncea*, *Brassica carinata* and *Brassica nigra*. ClustalW was used to align the full protein sequences of 76 cyclins, and MEGA X was used to create a phylogenetic tree using the neighbor-joining method with 1000 bootstrap repetitions. Based on the nomenclature of the *Arabidopsis* cyclins (from A-, B-, and D- plant-specific cyclins), all cyclins were divided into six unique groups and were differentiated by distinctive colors.

**Figure 4 ijms-23-14017-f004:**
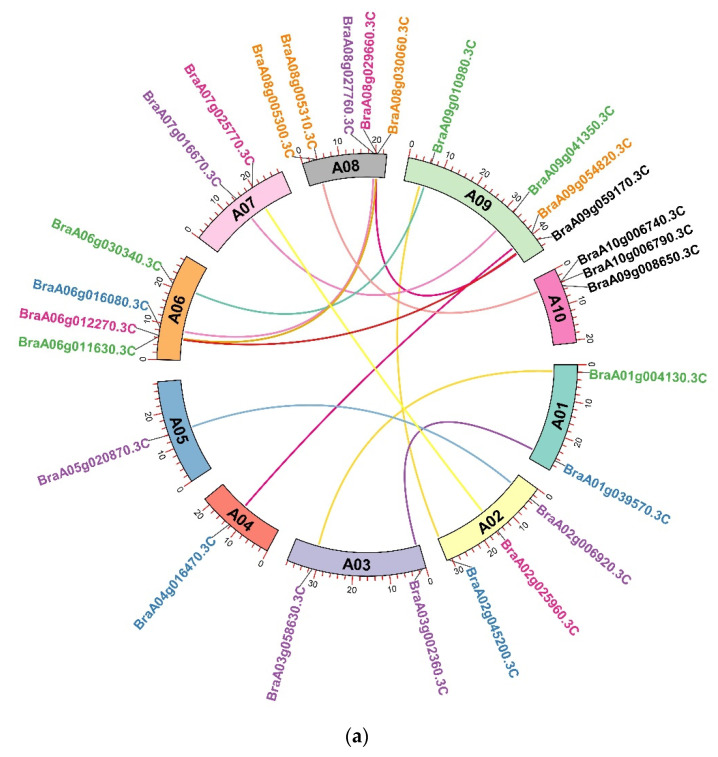
(**a**) Circos figure for the chromosome locations with the *Brcyclins* segmental duplication links. (**b**) *Brcyclin* synteny study in *Arabidopsis thaliana*, *Brassica napus*, *Brassica oleracea*, *Brassica juncea*, *Brassica carinata*, and *Brassica nigra*. The syntenic cyclin gene pairs between the mentioned species are indicated by black lines. The gray line in the backdrop symbolizes the collinear blocks.

**Figure 5 ijms-23-14017-f005:**
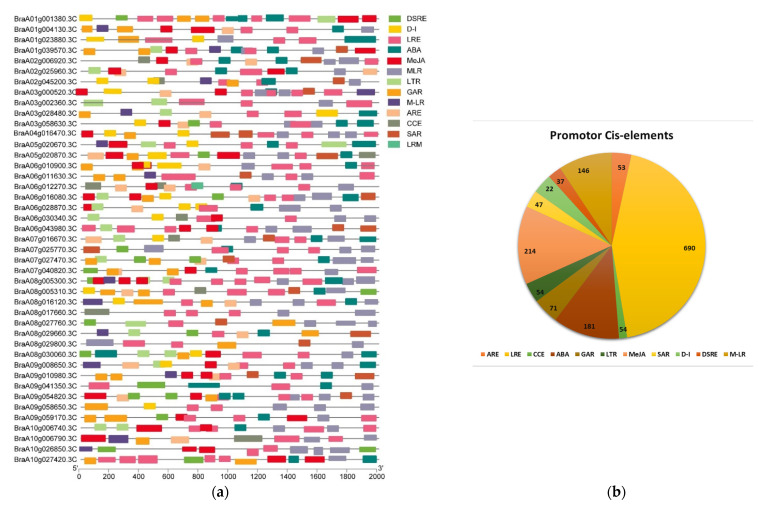
Identified cis-acting elements in the cyclin gene family. (**a**) cis-elements; Different hormone- and stress-responsive elements are linked to cis-elements in the promoters of the *Brcyclin* genes. Different colored boxes represent different cis-elements. (**b**) The pie chart shows the total identified elements and the number of each element in a specific color. Auxin responsive elements (ARE), light responsive elements (LRE), circadian responsive elements (CCE), abscisic acid (ABA), gibberellic acid-responsive elements (GA), low temperature-responsive elements (LTR), methyl jasmonate (MeJA), salicylic acid (SAR), drought inducible (D-I), defense and stress-responsive elements (DSRE), MYB binding site involved in light responsiveness (M-LR).

**Figure 6 ijms-23-14017-f006:**
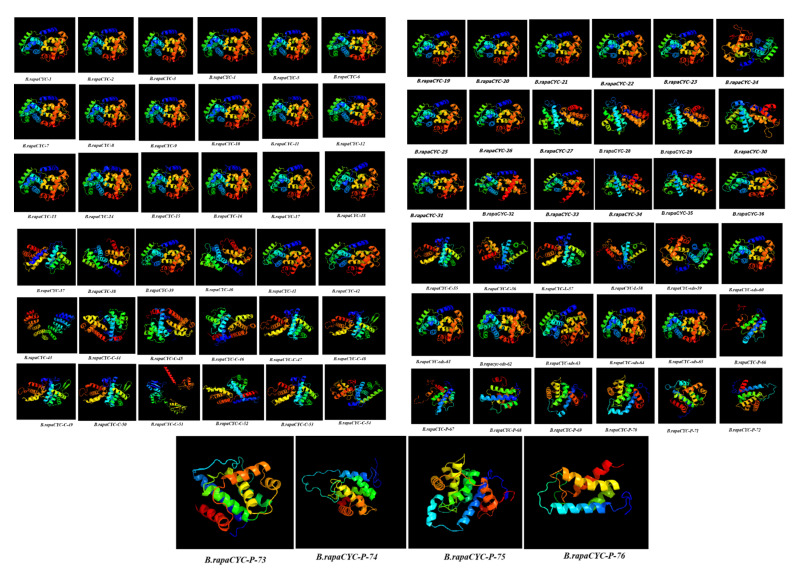
Predicted 3D-Structure of the cyclin gene family members in *Brassica rapa*.

**Figure 7 ijms-23-14017-f007:**
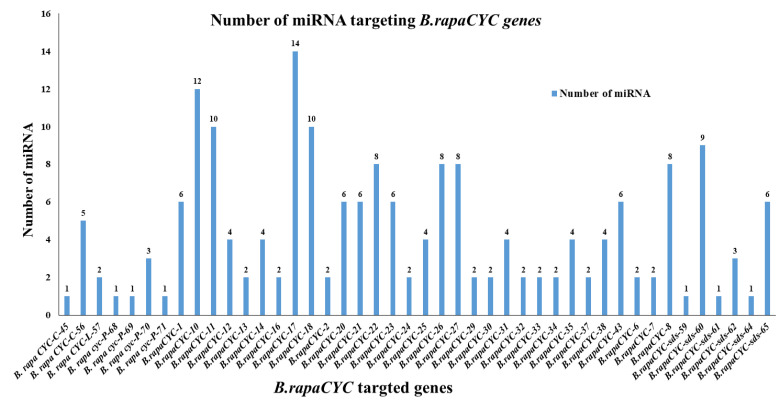
Graphical description of the miRNA targeting cyclin genes in *Brassica rapa*.

**Figure 8 ijms-23-14017-f008:**
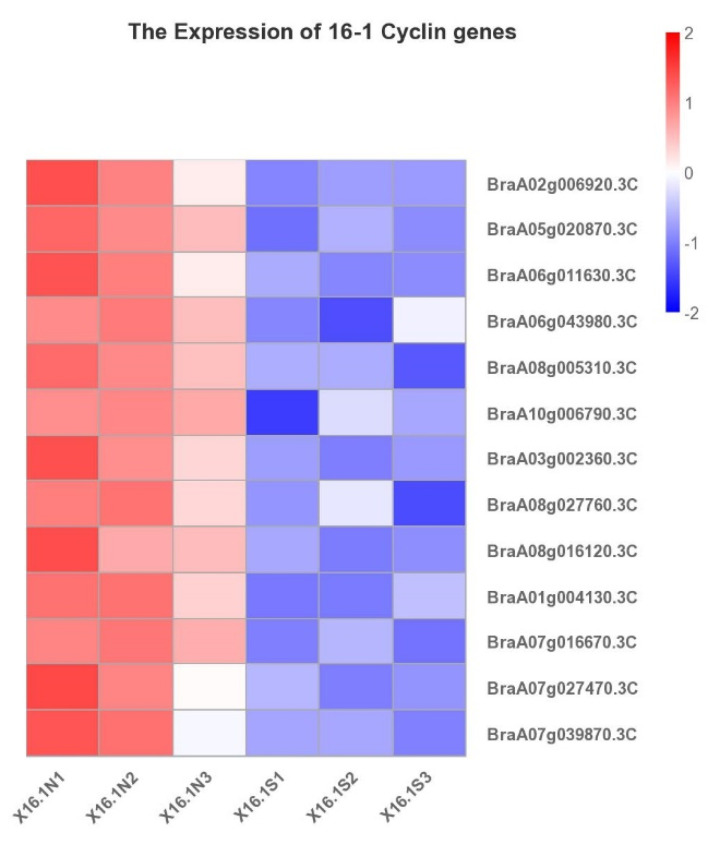
Transcriptomic expression analysis of the *Brcyclin* genes in the Chinese cabbage-cabbage translocation line 16-1. |**log_2_FC**| ≥ 1, FDR ≤ 0.01.

**Figure 9 ijms-23-14017-f009:**
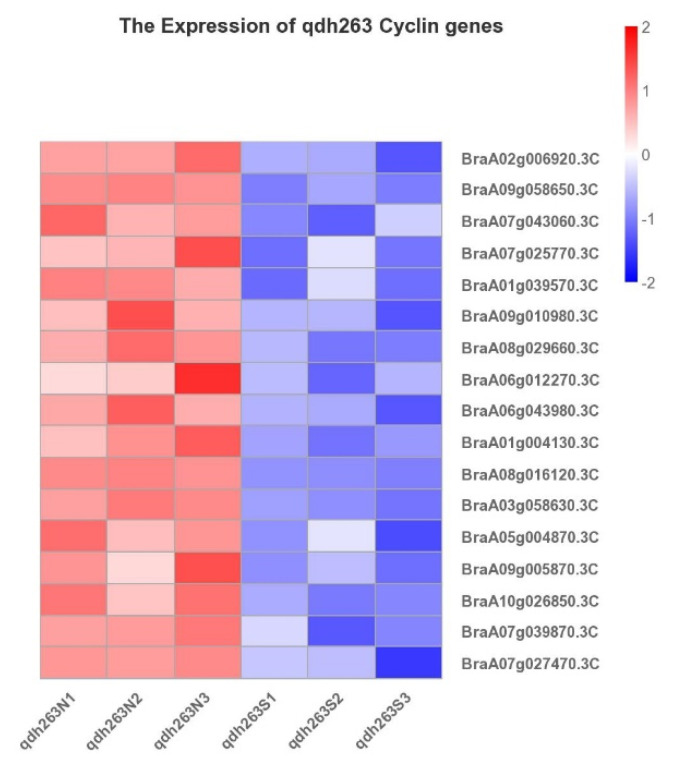
Transcriptomic expression analysis of the *Brcyclin* genes in the Chinese cabbage-cabbage translocation line qdh-263. |**log_2_FC**| ≥ 1, FDR ≤ 0.01.

**Figure 10 ijms-23-14017-f010:**
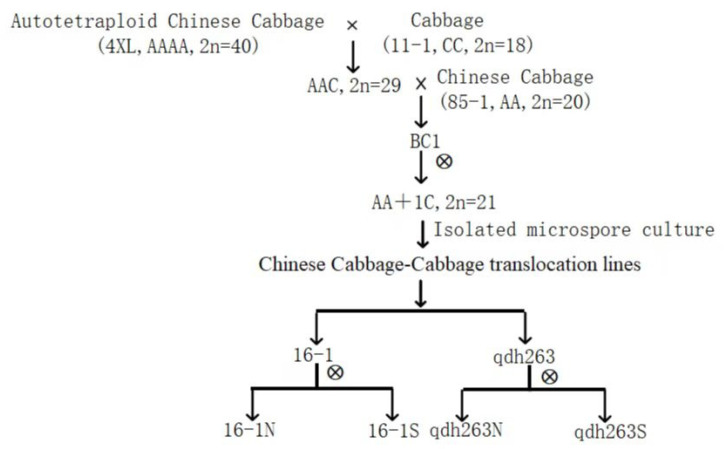
Source of research materials.

**Figure 11 ijms-23-14017-f011:**
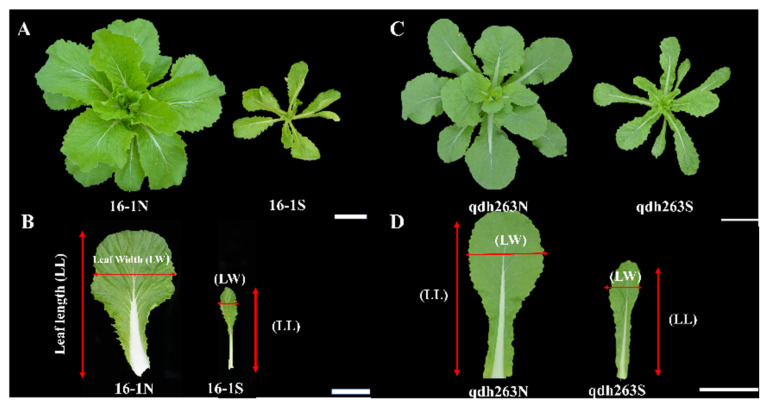
Chinese cabbage-cabbage translocation lines. (**A**) 16-1 translocation line; (**B**) leaves from 16-1 translocation line Normal (N) verses Small (S), (**C**) qdh-263 translocation line (**D**) qdh-263 translocation line leaves Normal (N) vs. Small (S).

**Table 1 ijms-23-14017-t001:** List of the cyclin genes identified in *Brassica rapa*. Predicted genes and related information.

Transcript ID	Gene Name	Given Name	Chr Position	Location Start-End	Strand	CDS (BP)	Instability Index	Amino Acid	PI	GRAVY	Intron, Exon	Sub-Cellular Localization	Molecular Weight
BraA05g020870.3C	CYCA1;1; CYCLIN A1;1	*B. rapaCYC-1*	A05	13,877,912–13,880,523	Positive	1332	54.22	443	6.06	−0.357	9, 10	Nuclear	50,140.08
BraA02g006920.3C	CYCA1;1; CYCLIN A1;1	*B. rapaCYC-2*	A02	3,270,499–3,272,381	Positive	1281	56.27	426	8.23	−0.345	7, 8	Nuclear	47,913.92
BraA07g040820.3C	CYCA1; CYCA1;2; CYCLIN A1; CYCLIN A1;2; DYP; TAM; TARDY ASYNCHRONOUS MEIOSIS	*B. rapaCYC-3*	A07	27,632,105–27,634,004	Negative	1161	54.03	386	7.5	−0.288	9, 10	Nuclear	43,398.62
BraA10g026850.3C	-	*B. rapaCYC-4*	A10	17,761,806–17,764,332	Positive	1314	53	437	5.71	−0.314	11, 12	Nuclear	49,155.22
BraA06g011630.3C	CYCA2;3; CYCLIN A2;3	*B. rapaCYC-5*	A06	6,250,506–62,52,628	Positive	1311	46.13	436	8.01	−0.256	9, 10	Nuclear	49,186.32
BraA03g028480.3C	CYCA2;4; CYCLIN A2;4	*B. rapaCYC-6*	A03	14,260,805–14,262,519	Positive	1134	44.44	377	8.15	−0.214	7, 8	Nuclear	43,351.16
BraA09g058650.3C	CYCA2;3; CYCLIN A2;3	*B. rapaCYC-7*	A03	41,527,085–41,528,756	Negative	1230	46.23	409	8.77	−0.336	6, 7	Nuclear	46,702.59
BraA02g025960.3C	CYCA2;4; CYCLIN A2;4	*B. rapaCYC-8*	A02	15,596,329–15,598,477	Negative	1374	42.77	457	8.83	−0.223	8, 9	Nuclear	51,107.63
BraA08g029800.3C	CYCA2;3; CYCLIN A2;3	*B. rapaCYC-9*	A08	20,411,681–20,413,662	Negative	1323	39.87	440	8.41	−0.278	8, 9	Extracellular	49,730.95
BraA07g025770.3C	CYCA2;4; CYCLIN A2;4	*B. rapaCYC-10*	A07	19,794,313–19,796,231	Positive	1371	44.44	456	9.01	−0.23	6, 7	Nuclear	51,266.89
BraA08g005300.3C	CYCA3;4; CYCLIN A3;4	*B. rapaCYC-11*	A08	4,068,714–4,070,324	Negative	1094	49.99	364	8.49	−0.295	6, 7	Nuclear	41,532.04
BraA10g006790.3C	CYCA3;2; CYCLIN-DEPENDENT PROTEIN KINASE 3;2	*B. rapaCYC-12*	A10	3,874,537–3,876,232	Negative	1092	49.99	363	8.81	−0.235	6, 7	Nuclear	41,697.52
BraA08g005310.3C	CYCA3;2; CYCLIN-DEPENDENT PROTEIN KINASE 3;2	*B. rapaCYC-13*	A08	4,072,383–4,074,097	Negative	1092	50.24	363	8.82	−0.202	6, 7	Nuclear	41,530.37
BraA05g020670.3C	CYCA3;2; CYCLIN-DEPENDENT PROTEIN KINASE 3;2	*B. rapaCYC-14*	A05	13,592,971–13,594,670	Positive	1074	52.74	357	8.80	−0.223	6, 7	Nuclear	40,842.66
BraA10g006740.3C	CYCA3;4; CYCLIN A3;4	*B. rapaCYC-15*	A10	3,850,757–3,852,385	Negative	1116	44.50	371	8.22	−0.156	6, 7	Nuclear	42,301.20
BraA06g043980.3C	CYCA3;1; CYCLIN A3;1	*B. rapaCYC-16*	A06	28,643,352–28,644,921	Negative	1062	41.28	353	9.08	−0.271	5, 6	Nuclear	41,001.89
BraA06g016080.3C	CYCB2;3; CYCLIN B2;3	*B. rapaCYC-17*	A06	8,732,392–8,734,446	Negative	1233	56.15	410	5.10	−0.360	9, 10	Cytoplasmic	46,910.55
BraA01g039570.3C	CYC2; CYCB1;3; CYCLIN 2; CYCLIN B1;3	*B. rapaCYC-18*	A01	26,658,540–26,660,047	Positive	1182	45.64	393	8.59	−0.183	4, 5	Mitochondrial	44,061.81
BraA03g002360.3C	CYC1BAT; CYCB1;2; CYCLIN B 1;2	*B. rapaCYC-19*	A03	1,051,126–1,052,853	Negative	1245	49.12	414	9.28	−0.278	4, 5	Mitochondrial	46,480.05
BraA07g027470.3C	-	*B. rapaCYC-20*	A07	20,732,255–20,734,238	Positive	1176	53.54	391	6.57	−0.332	9, 10	Cytoplasmic	44,996.94
BraA08g027760.3C	CYCB2;3; CYCLIN B2;3	*B. rapaCYC-21*	A08	19,536,921–19,538,864	Positive	1305	54.51	434	5.07	−0.302	8, 9	Cytoplasmic	49,415.67
BraA06g030340.3C	CYCB2;1; CYCLIN B2;1	*B. rapaCYC-22*	A06	20,983,296–20,985,215	Negative	1203	45.41	400	5.17	−0.228	7, 8	Mitochondrial	45,966.08
BraA09g010980.3C	CYCB2;1; CYCLIN B2;1	*B. rapaCYC-23*	A09	6,317,709–6,319,632	Positive	1230	47.67	409	4.99	−0.229	8, 9	Nuclear	46,902.91
BraA01g001380.3C	CYC1; CYCB1; CYCB1;1; CYCLIN 1; CYCLIN B1;1	*B. rapaCYC-24*	A01	691,860–693,690	Positive	1443	50.48	480	8.65	−0.285	5, 6	Nuclear	54,700.26
BraA06g012270.3C	CYCB3;1; CYCLIN B3;1	*B. rapaCYC-25*	A06	6,551,850–6,555,992	Negative	1758	38.81	585	9.81	−0.529	19, 20	Nuclear	65,590.43
BraA08g029660.3C	CYCB3;1; CYCLIN B3;1	*B. rapaCYC-26*	A08	20,349,587–20,354,332	Positive	1962	41.35	653	9.89	−0.550	20, 21	Nuclear	73,070.79
BraA08g030060.3C	-	*B. rapaCYC-27*	A08	20,500,274–20,502,784	Positive	1593	60.11	530	5.67	−0.537	6, 7	Nuclear	60,382.71
BraA06g010900.3C	-	*B. rapaCYC-28*	A06	5,915,727–5,918,238	Negative	1542	55.82	513	5.73	−0.416	5, 6	Nuclear	57,936.04
BraA09g059170.3C	-	*B. rapaCYC-29*	A09	41,774,847–41,777,896	Positive	1497	64.00	498	5.09	−0.487	6, 7	Nuclear	56,254.84
BraA10g027420.3C	CYCD4;2; CYCLIN D4;2	*B. rapaCYC-30*	A10	18,020,230–18,021,446	Positive	771	53.86	256	5.48	−0.139	5, 6	Plasma Membrane	29,511.92
BraA09g054820.3C	CYCD2;1; CYCLIN D2;1	*B. rapaCYC-31*	A09	39,657,751–39,659,377	Positive	1068	49.28	355	5.18	−0.426	5, 6	Nuclear	39,959.02
BraA09g008650.3C	-	*B. rapaCYC-32*	A10	4,933,003–4,934,800	Negative	852	51.80	283	5.2	−0.213	5, 6	Cytoplasmic	32,345.25
BraA04g016470.3C	CYCD2;1; CYCLIN D2;1	*B. rapaCYC-33*	A04	12,503,632–12,505,304	Negative	1071	52.72	356	5.1	−0.417	5, 6	Nuclear	40,100.31
BraA06g028870.3C	-	*B. rapaCYC-34*	A06	20,023,516–20,025,539	Negative	933	54.78	310	5.06	−0.086	5, 6	Nuclear	35,054.22
BraA02g045200.3C	-	*B. rapaCYC-35*	A02	31,028,998–31,030,868	Negative	900	49.48	299	5.61	−0.030	5, 6	Plasma Membrane	33,654.77
BraA08g017660.3C	-	*B. rapaCYC-36*	A08	14,172,516–14,173,638	Negative	771	37.25	256	4.56	0.030	4, 5	Extracellular	28,879.19
BraA07g016670.3C	-	*B. rapaCYC-37*	A07	14,617,049–14,618,272	Negative	1065	65.29	354	5.03	−0.25	2, 3	Nuclear	40,955.86
BraA03g000520.3C	-	*B. rapaCYC-38*	A03	229,865–231,583	Positive	1032	47.01	343	5.38	−0.13	4, 5	Cytoplasmic	39,248.30
BraA03g058630.3C	CYCD3; CYCD3;1; CYCLIN D3;1	*B. rapaCYC-39*	A03	31,000,883–31,002,215	Positive	1107	71.49	368	4.91	−0.327	3, 4	Nuclear	42,248.99
BraA09g041350.3C	CYCD3;3; CYCLIN D3;3	*B. rapaCYC-40*	A09	32,248,723–32,250,058	Negative	1062	72.29	353	4.77	−0.225	3, 4	Nuclear	40,740.21
BraA01g004130.3C	CYCD3; CYCD3;1; CYCLIN D3;1	*B. rapaCYC-41*	A01	2,002,141–2,003,508	Negative	1122	68.86	373	4.77	−0.279	3, 4	Nuclear	42,422.18
BraA08g016120.3C	CYCD3; CYCD3;1; CYCLIN D3;1	*B. rapaCYC-42*	A08	13,174,233–13,175,588	Negative	1107	62.40	368	4.94	−0.249	3, 4	Nuclear	42,223.12
BraA01g023880.3C	-	*B. rapaCYC-43*	A01	13,763,901–13,768,717	Positive	3057	41.82	1018	8.59	−0.246	6, 6	Nuclear	113,983.17
BraA02g039780.3C	-	* B. rapaCYC-C-44 *	A02	27,675,140–27,676,606	Negative	762	42.96	253	6.6	−0.165	7, 8	Mitochondrial	29,767.61
BraA06g035450.3C	-	* B. rapaCYC-C-45 *	A06	23,940,789–23,942,556	Positive	759	41.16	252	5.8	−0.158	8, 9	Cytoplasmic	29,694.42
BraA02g039790.3C	-	* B. rapaCYC-C-46 *	A02	27,680,640–27,682,190	Negative	693	41.82	230	6.14	0.011	7, 8	Plasma Membrane	27,030.48
BraA01g010840.3C	-	* B. rapaCYC-C-47 *	A01	5,623,479–5,625,573	Positive	1218	48.66	405	4.92	−0.57	5, 6	Nuclear	45,152.52
BraA02g031590.3C	-	* B. rapaCYC-C-48 *	A02	21,630,856–21,633,310	Negative	1680	52.32	559	6.8	−0.885	5, 6	Nuclear	62,360.51
BraA08g013560.3C	CYCT1;4	* B. rapaCYC-C-49 *	A08	11,600,082–11,602,744	Positive	1584	46.85	527	6.1	−0.867	5, 6	Nuclear	58,732.46
BraA06g042650.3C	-	* B. rapaCYC-C-50 *	A06	27,936,496–27,939,199	Positive	1704	53.51	567	6.72	−1.012	5, 6	Nuclear	63,905.01
BraA01g010870.3C	-	* B. rapaCYC-C-51 *	A01	5,633,272–5,640,778	Positive	3690	49.3	1229	7.03	−0.71	20, 21	Nuclear	138,166.86
BraA09g005110.3C	-	* B. rapaCYC-C-52 *	A09	3,048,451–3,050,956	Positive	1005	42.03	334	7.24	−0.54	12, 13	Nuclear	37,950.23
BraA03g011410.3C	CYCLIN T1;1; CYCT1;1	* B. rapaCYC-C-53 *	A03	4,938,992–4,939,786	Positive	795	53.3	264	8.37	−0.385	0, 1	Cytoplasmic	30,399.98
BraA08g024790.3C	-	* B. rapaCYC-C-54 *	A08	18,035,515–18,037,466	Negative	963	59.56	320	8.83	−0.37	5, 6	Nuclear	37,298.98
BraA07g012460.3C	ARGININE-RICH CYCLIN 1; ATRCY1; CYCL1; MODIFIER OF SNC1 12; MOS12; RCY1	* B. rapaCYC-C-55 *	A07	11,913,217–11,915,226	Negative	1242	56.16	413	9.08	−0.883	5, 6	Nuclear	47,528.46
BraA07g012040.3C	-	* B. rapaCYC-C-56 *	A07	11,699,847–11,702,109	Negative	942	57.18	313	8.1	−0.294	5, 6	Plasma Membrane	36,468.93
BraA09g037340.3C	ARGININE-RICH CYCLIN 1; ATRCY1; CYCL1; MODIFIER OF SNC1 12; MOS12; RCY1	* B. rapaCYC-L-57 *	A09	29,511,874–29,517,827	Negative	1329	40.29	442	4.78	−0.241	6, 7	Cytoplasmic	50,159.53
BraA01g028190.3C	-	*B. rapaCYC-L-58*	A01	18412523–18413938	Positive	795	54.52	264	4.87	−0.368	4, 5	Nuclear	29,472.28
BraA02g001920.3C	CYC1BAT; CYCB1;2; CYCLIN B 1;2	*B. rapaCYC -sds-59*	A02	965,051–966,455	Negative	1230	47.41	409	9.04	−0.184	2, 3	Mitochondrial	45,404.66
BraA07g039870.3C	-	*B. rapaCYC-sds-60*	A07	27,174,809–27,176,858	Negative	1296	54.9	431	5.02	−0.414	9,10	Cytoplasmic	49,141.14
BraA03g059620.3C	CYCB2;2; CYCLIN B2;2	*B. rapaCYC -sds-61*	A03	31,664,889–31,666,889	Positive	1254	62.85	417	4.88	−0.325	7, 8	Nuclear	48,241.18
BraA09g005870.3C	-	*B. rapaCYC-sds-62*	A09	3,427,999–3,430,021	Positive	1329	45.06	442	9.26	−0.379	7, 8	Nuclear	50,217.46
BraA05g030530.3C	CYCA1; CYCA1;2; CYCLIN A1; CYCLIN A1;2; DYP; TAM; TARDY ASYNCHRONOUS MEIOSIS	*B. rapaCYC-sds-63*	A05	22,429,978–22,431,111	Positive	1134	53.68	377	8.13	−0.277	0, 1	Nuclear	42,650.81
BraA07g043060.3C	CYCA2;4; CYCLIN A2;4	*B. rapaCYC-sds-64*	A07	28,644,746–28,646,946	Negative	1392	46.72	463	9.07	−0.237	8, 9	Nuclear	52,063.95
BraA03g023450.3C	CYCLIN P4;1; CYCP4;1	*B. rapa CYC-P-65*	A03	11,242,736–11,243,586	Negative	600	46.36	199	6.19	0.014	1, 2	Plasma Membrane	22,646.98
BraA04g030980.3C	CYCLIN P4;1; CYCP4;1	*B. rapa CYC-P-66*	A04	21,178,657–21,179,672	Positive	606	49.62	201	6.96	0.011	1, 2	Plasma Membrane	22,884.32
BraA05g004620.3C	CYCLIN P4;1; CYCP4;1	*B. rapa CYC-P-67*	A05	2,346,286–2,347,847	Negative	612	52.23	203	5.62	−0.075	4, 5	Plasma Membrane	23,304.58
BraA10g029800.3C	CYCLIN P4;3; CYCP4;3	*B. rapa CYC-P-68*	A10	18,944,230–18,944,973	Positive	651	52.66	216	4.87	−0.09	1, 2	Plasma Membrane	24,974.67
BraA05g004870.3C	CYCLIN P3;1; CYCP3;1	*B. rapa CYC-P-69*	A05	2,493,781–2,494,542	Positive	669	37.86	222	8.53	−0.157	1, 2	Plasma Membrane	25,520.26
BraA09g050340.3C	CYCLIN P3;2; CYCP3;2	*B. rapa CYC-P-70*	A09	37,331,296–37,332,086	Positive	702	35.67	233	9.28	35.67	1, 2	Plasma Membrane	26,834.83
BraA01g042720.3C	-	*B. rapa CYC-P-71*	A01	28,475,151–28,475,963	Positive	639	60.56	212	9.14	−0.238	2, 3	Plasma Membrane	24,505.31
BraA03g023540.3C	CYCLIN P3;1; CYCP3;1	*B. rapa CYC-P-72*	A03	11,293,034–11,293,822	Positive	675	35.66	224	8.17	−0.162	1, 2	Plasma Membrane	25,723.51
BraA05g039890.3C	-	*B. rapa CYC-P-73*	A05	27,128,568–27,129,415	Positive	639	45.46	212	8.99	−0.12	2, 3	Plasma Membrane	24,389.31
BraA09g052400.3C	CYCLIN P1;1; CYCP1;1	*B. rapa CYC-P-74*	A09	38,262,874–38,263,612	Negative	668	38.92	222	8.71	−0.205	1, 2	Nuclear	25,347.16
BraA05g025300.3C	CYCLIN P2;1; CYCP2;1	*B. rapa CYC-P-75*	A05	19,023,223–19,024,100	Positive	633	31.84	210	5.67	0.077	1, 2	Extracellular	23,311.83
BraA03g044680.3C	CYCLIN P4;2; CYCP4; CYCP4;2	*B. rapa CYC-P-76*	A03	22,557,276–22,557,617	Positive	342	51.97	113	8.6	0.065	0, 1	Plasma Membrane	13,112.27

**Table 2 ijms-23-14017-t002:** Ka/Ks value of the *Brassica rapa* cyclin genes.

Seq_1	Seq_2	Ka	Ks	Ka/Ks	(TMYA)
*B. rapaCYC-5*	*B. rapaCYC-7*	0.07466907	0.298777734	0.249915108	9,959,257.814
*B. rapaCYC-8*	*B. rapaCYC-10*	0.072044272	0.220167778	0.327224413	7,338,925.922
*B. rapaCYC-1*	*B. rapaCYC-2*	0.060408096	0.192325642	0.314092783	6,410,854.739
*B. rapaCYC-11*	*B. rapaCYC-15*	0.055887039	0.373769141	0.149522882	12,458,971.38
*B. rapaCYC-12*	*B. rapaCYC-13*	0.071777045	0.312268577	0.229856766	10,408,952.56
*B. rapaCYC-18*	*B. rapaCYC-19*	0.197132861	1.12897999	0.174611474	37,632,666.33
*B. rapaCYC-25*	*B. rapaCYC-26*	0.081638479	0.254689097	0.320541713	8,489,636.554
*B. rapaCYC-22*	*B. rapaCYC-23*	0.064868387	0.313106866	0.207176506	10,436,895.54
*B. rapaCYC-17*	*B. rapaCYC-21*	0.063775726	0.283464628	0.224986542	9,448,820.944
*B. rapaCYC-27*	*B. rapaCYC-29*	0.110497066	0.246929994	0.44748337	8,230,999.811
*B. rapaCYC-39*	*B. rapaCYC-41*	0.057811503	0.335051252	0.172545253	11,168,375.06
*B. rapaCYC-37*	*B. rapaCYC-40*	0.217039312	0.789508716	0.274904264	26,316,957.2
*B. rapaCYC-31*	*B. rapaCYC-33*	0.061048457	0.297976348	0.204876854	9,932,544.937
*B. rapaCYC-32*	*B. rapaCYC-35*	0.137015441	0.260078921	0.526822552	8,669,297.358

**Table 3 ijms-23-14017-t003:** Gene annotation of the differentially expressed cyclin genes.

*Brassica rapa* Gene ID	*Arabidopsis* Gene ID	Gene Name	Gene Annotation
BraA02g006920.3C	AT1G44110	*CYCA1*	Cyclin A1
BraA05g020870.3C	AT1G44110	*CYCA1;1*	Cyclin A1
BraA06g011630.3C	AT1G15570	*CYCA2;3*	A2-type cyclin. In the *Arabidopsis* endoreduplication, it negatively affects the endocycles and acts as a key regulator of the ploidy levels.
BraA06g043980.3C	AT5G43080	*CYCA3;1*	Cyclin A3
BraA08g005310.3C	AT1G47210	*CYCA3;2*	cyclin-dependent protein kinase 3
BraA10g006790.3C	AT1G47210	*CYCA3;2*	cyclin-dependent protein kinase 3
BraA03g002360.3C	AT5G06150	*CYCB1;2*	Encodes a cyclin whose expression is reduced in response to high salt
BraA08g027760.3C	AT1G20610	*CYCB2;3*	Cyclin B2
BraA08g016120.3C	AT4G34160	*CYCD3;1*	Encodes a cyclin D-type protein. Plays its role in the transition from the cell proliferation to the final phases of differentiation. Regulated by Cytokinin and brassinosteroid.
BraA01g004130.3C	AT4G34160	*CYCD3;1*	Encodes a cyclin D-type protein. Function in shift from the cell proliferation to final stages of differentiation.
BraA07g016670.3C	AT5G67260	Uncharacterized Gene	Encodes CYCD3;2 a CYCD3 D-type cyclin. Important for controlling cell numbers in developing organs and for modulating the effects of the cytokinins in the apical growth and development. With PPD and NINJA, plays important role in the leaf morphogenesis conjunction.
BraA07g027470.3C	AT1G76310	Uncharacterized Gene	Core cell cycle genes
BraA07g039870.3C	AT1G76310	Uncharacterized Gene	Core cell cycle genes

## Data Availability

The datasets used and/or analyzed during the current study are available from the corresponding author upon reasonable request. However, most of the data is shown in the Appendix A.

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
