# Peer review of "Genome-Wide Identification, Characterization, and Transcriptomic Analysis of the Cyclin Gene Family in Brassica rapa"

_ijms, 2022, doi:10.3390/ijms232214017_

Round 1

Reviewer 1 Report

The article submitted presents the genomic characterization of cyclin gene family in Brassica rapa species, which plays a critical role in cell proliferation by activating crucial enzymes. The research presented in this article provides some useful insights into the cyclin gene family. However, the article needs to be revised substantially.

Comments:

·       Line 67, remove “Aside from A and B type cyclins”

·       The introduction mostly discusses A and B-type cyclins and hasn’t given any information on the E-type cyclins. Moreover, why the author choose to study this topic should be strengthened

·       The introduction needs to be shortened and carefully revised for grammar and scientific writing mistakes.

·       On what basis only 76 transcripts were kept?

·       Line 49 and line 129 present the same information and redundant information should be deleted from the whole manuscript.

·       In the results, the authors have cited Table 2 before table 1.

·       Figure 6 indicates predicts the protein structures, which do not add useful information to the article. I would recommend adding figure 6 in supplementary files or briefly adding a relevant explanation of protein structure.

·       Which program of BLAST (blastp or blastn) is used during the identification of genes along with the accessed date?

·       Which sequence of cyclin gene is used during GSDS analysis should mention (either coding sequence or DNA)

·       How authors have identified the domains of cyclin genes (which tool like SMART or NCBI CD search, here MEME is mentioned but MEME uses for only MOTIF analysis.

·       Throughout the article, the authors have used different writing styles even for Arabidopsis, sometimes italic and sometimes nonitalic.

·       Discussion is redundant with information about the other cyclins gene in the other plant species, I would recommend to be pertinent to the role of genes in B.rapa.

·       Overall scientific writing needs and uniformity of text is needed to be checked carefully throughout the manuscript.

·       There is no uniformity of name in the syntenic diagrams.

Author Response

Dear reviewer, we would like to express our deep gratitude for the invaluable comments that helped us revise our manuscript. We are in the process of addressing, revising, and submitting our manuscript. All of the concerns and suggestions you raised have now been addressed.

Reviewer 2 Report

This nice work will help us understand the evolutionary functions of cyclins in Brassica rapa and other plant species. However, the author has touched on several points in this manuscript, making it hard to grasp it as a story. Therefore, I suggest that the authors consider the following points to make it a coherent story.

1. There are ten types of cyclins in Arabidopsis based on function and sequence analysis; how did the author decide which cyclin to use for BLASTP to retrieve Brassica Rapa cyclins? How did author find out that a few cyclin types (A-, B-, D-, and P-) were further subdivided into several subtypes, not others (C-, H-, L-, SDS-, and T-)? Is there any specific characteristic of cyclin types, or what is the basis of cyclin classification?

2. Line-107, P-type had different genes. What is this mean?

3. Please describe in detail (in text or materials and method) how 76 distinct cyclin genes are identified from Brassica rapa. I am not convinced.

4. In the result section 2.2, lines 127 and 128 are unclear to me. All cyclin members have the same gene structure, so does it mean all cyclins have the same number and type of motifs? These two lines are not consistent with rest of the paragraph.

5. In the result section 2.6, does author want to claim that all cyclins have similar functions based on the conserved 3D structure?

6. How does section 2.7 (miRNA targeting genes) fit in this story?

7. What is the take-home message from section 2.10 (Expression Profiles of Brassica rapa Cyclin Genes by Transcriptome Analysis)? It is explained in the discussion section; please elaborate in the result section as well.  

Author Response

(The authors gave the same response as above.)

Round 2

Reviewer 1 Report

Authors have addressed all of my concerns and it can be accepted for publication. 

Reviewer 2 Report

Thank you very much for addressing my suggestions. The manuscript seems ready for publication. Congrats to all authors.